# Fluid Flow in Continuous Casting Mold for Ultra-Wide Slab

**DOI:** 10.3390/ma16031135

**Published:** 2023-01-28

**Authors:** Gang Li, Lingfeng Tu, Qiangqiang Wang, Xubin Zhang, Shengping He

**Affiliations:** College of Materials Science and Engineering, and Chongqing Key Laboratory of Vanadium–Titanium Metallurgy and Advanced Materials, Chongqing University, Chongqing 400044, China

**Keywords:** ultra-wide slab, continuous casting mold, water model, cast speed, surface velocity

## Abstract

Ultra-wide slabs have a good application market and prospect, but there is still a lack of research on the flow field. To explore the characteristics of its flow field, this study built a 0.5-scale physical model of mold using Perspex. The effect of casting speed on flow field and surface flow speed was investigated by using an ink tracer experiment and contact measurement. There were various flow patterns in the ultra-wide slab mold, and they continue to transform each other. The jet momentum from the nozzle ports was diffused by colliding with the wide face, which lowered its kinetic energy and affected its subsequent diffusion. Compared with the conventional mold, the upper flow intensity of the ultra-wide slab mold was weaker, which made its liquid surface inactive and caused it to produce a flux rim or slag strip. At different casting speeds, the average flow speed distribution of the ultra-wide slab was C-shaped. When it increased from 0.9 to 1.4 m/s, the corresponding maximum average flow speed increased from 0.08 to 0.2 m/s. At the same time, the proportion of the low-flow speed zone at the most active part of the surface also gradually decreased from more than 90% to about 49%.

## 1. Introduction

With the maturity of slab continuous casting (CC) technology, the process has been pursuing ultra-high speed, ultra-thick and ultra-wide slab production. Usually, slabs with widths of above 1600 mm are regarded as wide slabs [1,2]. Wide slabs have the advantages of safety and integrity as well as less welding, and thus has been widely used in the large ocean engineering and shipbuilding industry, large bridges, large caliber oil and natural gas pipeline and other fields [3,4].

From the first wide slab CC of the Kawasaki Steel Plant in Japan in the 1970s to the present, many scholars have conducted a lot of research and have contributed to the development of wide slab production technology [5,6,7]. Kalter et al. [8] studied the effect of the width-to-thickness ratio on the fluctuation of fluid flow in slab molds. It was found that when the width-to-thickness ratio was 18, the molten steel first impacted the wide surface after being injected from the nozzle, and two vortices were formed in the upper flow. When the width-to-thickness ratio was 6.5, the steel jet punched straight into the narrow surface, forming upper and lower roll flows. Gupta et al. [9] studied the effects of typical nozzles and port angles on the flow field and jet characteristics in a 1600 × 70 mm slab mold by using water models. The results showed that the jet characteristics changed with the exit angle of the nozzle. As the angle was upward or horizontal, the jet presented a rotating stream shape, while the stream presented a smooth stream shape with a downward angle. By establishing a 1:1 water model, Meng et al. [10] studied the effects of different factors on the flow field, such as the number of ports, the air flow rate, the submerged depth of the nozzle, the casting speed and the cross-section of the mold. It was found that the fluctuation of surface flow with a three-hole nozzle was about 25% lower than that with two-hole nozzle, but the impact depth of the bottom-hole stream was 50% deeper than that of the side-hole stream. In addition, the surface flow fluctuation increased with the increase in gas flow rate and casting speed, and the reasonable casting speed range was different with casting different widths of strands. At present, the research on the flow field of wide slabs is mostly focused on ordinary conditions with slab widths below 2500 mm [10,11,12,13], while related research on the flow field of ultra-wide slabs with widths greater than 2500 mm is still rare.

Due to the increased flow rate of steel in wide molds compared with ordinary slabs, it is easy to lead to an unreasonable flow state of molten steel, which causes the temperature field in the mold to be more uneven and even affects the quality of the slab. The wider the slab, the greater the shell bending stress and condensation shrinkage force, and the more prone the strand is to longitudinal cracks, transverse cracks and internal segregation [14,15,16]. Figure 1 shows the condition of an ultra-wide mold with a cross-section of 2920 × 150 mm in a steel plant. The top surface of the slag pool was more active near the narrow side, while there were many solidified slag clusters at other areas. Longitudinal cracks often appeared near the center of the broad surface on the strand, which may be related to the poor inflow of the liquid slag and uneven cooling of the slab surface. To solve the above problems, the properties of the mold slag were optimized; however, these measures did not work. The analysis suggests that these problems may be caused by the weak upper circulation flow in the ultra-wide mold. The flow field determines the distribution of the temperature field, and the local low temperature zone appears due to the slow molten steel renewal at the steel–slag interface. Thus, the melting and flow of the liquid slag was not smooth, and solidified clusters were easily formed at the top surface, which would greatly increase the occurrence of longitudinal cracks, as well as sticker and breakout [17,18,19,20].

Therefore, it is of great significance to study the characteristics of ultra-wide slab flow fields for the development of the steel industry. In this study, the flow field in a mold with a cross-section of 2920 × 150 mm was determined by a physical model, and the influences of casting speed on the flow field and surface flow speed were investigated, which provided a reference for the production and technical development of ultra-wide slab continuous casting.

## 2. Experimental Methodology

A physical model is an important means to study flow field characteristics in a mold. It is simple and safe to operate, and the evolution of the main stream in the mold can be easily observed. A 0.5-scale water model was built based on the similarity principle, and the Froude numbers were equaled between the water model and the prototype. For single-phase fluid flow, fluid in the mold is mainly affected by the inertia force, viscous force and gravity [21]. Therefore, both the model and the prototype needed to satisfy the Reynolds numbers and the Froude numbers. Because the kinematic viscosity of water and molten steel is very close, the motion behavior of them are very similar. When the flow in the mold is completely turbulent, the effect of the Reynolds numbers is negligible, only to equal the Froude number [22,23,24].
(1)$$Fr_{p}=Fr_{m}$$
(2)$$\frac{gl_{p}}{u_{p}^{2}}=\frac{gl_{m}}{u_{m}^{2}}$$
where the subscript *p* represents the prototype, m represents the model, *g* is the gravitational acceleration (m·s^−2^), *l* is the characteristic length (m), and *u* is the fluid velocity (m·s^−1^). The casting speed pull ratio of the prototype and model was obtained as $\sqrt{2}$:1.

The mold and nozzle models were made of organic glass. In order to accommodate the flat shape of the inner cavity of the mold, the lower part of the nozzle was made flat (Figure 2). The water model was a closed circulation system, the position of the liquid level was controlled using stopper rod, and the casting speed was controlled by a water pump and turbine flowmeter. The main parameters of the model and prototype are shown in Table 1. Figure 2 shows the experimental schematic diagram. In this study, water was selected to simulate molten steel, and ink was used as a tracer to illustrate the flow pattern of molten steel in the mold. The movement trajectory of ink was recorded by a camera. The flow speed measure applied was a contact measuring instrument, including MiniWater6 Micro flow velocity sensor, DAM3152 data acquisition module, and a computer. In order to minimize the experimental error caused by the influence of the probe on the liquid flow field of the mold, the probe selected in this study was the minitype probe, whose size was φ11 × 15 mm, and its scope of application was 0.02~5.0 m/s.

## 3. Results and Discussion

### 3.1. Flow Pattern in the Ultra-Wide Slab Mold

During CC operation, the quality of the final product depends largely on the flow pattern of the molten steel in the mold [25,26,27]. If the symmetry on both sides is poor, it might cause the surface fluid to fluctuate violently and can easily cause slag entrainment. Therefore, it is necessary to explore the flow pattern in an ultra-wide slab mold.

Figure 3 shows some typical evolutions of the fluid flow by tracking the ink diffusion, when the casting speed was 1.3 m/min, the nozzle was submerged at a depth of 150 mm, and the nozzle outport angle was 15°. The casting speed and nozzle submerged depth described here are the values corresponding to the prototype. It was found that under the same condition, various patterns existed, consisting of symmetrical flow (Figure 3a) and asymmetric flow (Figure 3b,c). For the current ultra-wide slab, the pattern of the flow field was always changing, and the three forms in Figure 3 would continue to transform each other. Even if this was normal symmetrical flow, the upper reflux was relatively weak and the surface was relatively calm.

As the fluid flow was symmetrical in the mold, the jet angle was approximately 32–33° downward (Figure 4), which was roughly twice the nozzle downward inclination angle. Such a large jet angle would affect the turbulence energy, thus affecting subsequent diffusion of the stream. The range in the flow jet angle obtained in this study was similar to that of conventional slabs reported in the literature [28,29], but the upper reflow intensity in the ultra-wide slab mold was much weaker. Therefore, in order to better illustrate the evolution of the flow jet in the motion to the narrow side, Figure 4b presents the motion of the flow jet from top view. It shows that the stream touched both wide sides of the mold only 0.3 s after being jetted and reached the narrow face after 1.5 s. The jet momentum from the nozzle ports was diffused by colliding with the wide face of the mold, which lowered its kinetic energy. Meanwhile, this also indicated that the molten steel scoured the solidification front too strong at the wide sides, which would lead to uneven growth of the shell and induce defects such as longitudinal cracks on the strand surface.

In order to explore the influence of casting speed on flow pattern, multiple sets of ink tracing experiments were carried out by changing the water flow rate under the premise of keeping other experimental conditions unchanged. The statistical results are listed in Figure 5. The proportion of symmetric flow was about 13~29%, while the proportion of asymmetric stream was about 71~87%. Bernd [30] and Ren [2] studied the flow fields of molds with slab section sizes of 1800 × 220 mm and 2040 × 200 mm, respectively. It was found that the proportions of asymmetric flow in the flow fields were 6~11% and 14~31%, respectively. Gupta [31] found that when the aspect ratio is greater than 6.25, asymmetric flows dominate. The proportion of symmetrical flow in the flow field will decrease significantly with the increase in the width–thickness ratio of the mold section. The reasons for this situation were mainly as follows: (1) Different scholars have different evaluation standards for jet symmetry. (2) The conventional slab jet outlet that impacted the narrow surface of the motion path was shorter, the stream failed to fully diffuse, and thus, the symmetrical flow accounted for was larger. The motion path of the ultra-wide slab jet was longer, and the diffusion of the stream was more serious, even hitting the two sides of the wall in advance and making the flow pattern of the stream complex and changeable. Therefore, the proportion of the symmetrical flow in the mold of the ultra-wide slab was significantly lower than that of the conventional slab. Gupta [31] studied the effect of casting speed on the flow field of a conventional slab mold. When the casting speed was 0.5 m/min, the lower circulation was an asymmetric flow, with a strong left side and a weak right side. When the casting speed was increased to 1.5 m/min, the lower circulation became a symmetrical circulation. For the ultra-wide slab mold, when the casting speed exceeded a certain value, the symmetry of the stream would deteriorate. In short, the flow field in the ultra-wide slab mold at various casting speeds was basically the same when the fluid flow pattern in the mold was constantly changing, and it was asymmetric most of the time.

### 3.2. Distribution Flow Speed at Meniscus for the Ultra-Wide Slab Mold

During the CC process, if the level fluctuation of molten steel is too intense, the liquid flux would be involved in the molten steel, resulting in slag-related defects. However, if the top surface is too calm, this indicates that the intensity of the upper flow field is weak, which is not conducive to the melting of the mold slag and the lubrication of the solidifying shell [32]. Therefore, it is critical to investigate the characteristics of the fluid flow at the meniscus for ultra-wide slab molds.

In this study, the flow speed 20 mm below the meniscus at different positions was first measured, as the casting speed was 1.3 m/min and the nozzle submerged depth was 150 mm. Figure 6 exhibits that the variation of flow speed with time at the positions of 90, 360 and 630 mm from the width center. The variation of surface flow speed with time near both sides of the nozzle (Figure 6a) or near the narrow face (Figure 6c) was relatively symmetrical, and the maximum surface speed was 0.23 and 0.24 m/s, respectively. However, the variation trend of the flow speed with time on both sides near the quarter width was quite asymmetric (Figure 6b). In the initial period of the measurement, approximately 0–550 s, the left surface level was calm, and its flow speed became diminished and fluctuated in the range of 0.052~0.356 m/s, while the right level fluctuated with a large flow speed beneath the meniscus, 0.02~0.422 m/s over the same period. Then, the trend of the comparison at both sides gradually reversed. This transformation corresponds to a process during which the flow pattern in the mold first changed from asymmetrical (Figure 3b) to symmetrical (Figure 3a) and then changed to asymmetrical (Figure 3c).

In order to intuitively reflect the variation of the flow speed at the meniscus, this study divided these obtained data into two ranges: above 0.2 m/s and below 0.2 m/s. Figure 7 shows the statistical result of the two ranges at one side of the mold. Within 180 mm of the nozzle and 90 mm of the narrow surface, the flow surface was relatively inactive, and the surface flow speed was basically less than 0.2 m/s. At the middle area between the nozzle and the narrow face, the two ranges both accounted for a large proportion, indicating that the meniscus always varied dynamically and transformed between calm and active, which may also be the reason that there are more clusters on the slag surface in Figure 1. The meniscus most of the time is not active enough, resulting in slow updating of the high-temperature molten steel at the steel–slag interface, which is not conducive to the flow of the mold flux and lubrication of the strand. Moreover, it is easy to produce more slag strips and clusters, which is harmful to the smooth running of continuous casting and the surface quality of the slab.

Figure 8 shows the distribution of the average surface flow speed and the measurement locations. The data were the average of the measured data within 20 min. The surface flow speed on both sides of the nozzle was distributed basically and symmetrically. Within 300–500 mm from the width center, the flow surface was the most active, and the flow speed was the largest, at about 0.14–0.23 m/s. Near the SEN and near the narrow side, it was 0.06–0.09 and less than 0.09 m/s, respectively. During the current measurement period, the flow speed near the broad surface on the left side of the nozzle was larger than that at the center of the thickness. This was consistent with the observation of the ink tracking experiment in Figure 4b: the jet from the nozzle impact with the wide surface moved first to the narrow surface, and the turbulent kinetic energy of the flow field near the wide surface was larger.

Figure 9 compares the influence of casting speed on the distribution of average flow speed in the mid-thickness section of the mold, under the premise of keeping other experimental conditions the same. Similar to the liquid surface flow field of conventional slabs, the flow speed distribution of the ultra-wide slab was C-shaped, such that the flow rate increased first and then decreased from the nozzle to the narrow side at different casting speeds. When the casting speed was 0.9 m/min, the maximum flow speed was about 0.09 m/s, which appeared at the position 270 mm from the width center. As the casting speed was increased to 1.0–1.2 m/min, the maximum flow speed varied slightly (about 0.15 m/s), while its position tended to be close to the narrow surface, about 400–420 mm away from the width center. As the casting speed continued to increase, the maximum flow speed was 0.18 and 0.20 m/s, respectively, for 1.3 and 1.4 m/min. The steel flow rate of the ultra-wide slab reached 4.30 ton/min at 1.4 m/min. At the same casting speed, it has been reported that the steel flow rate of conventional slabs is about 2.85~3.10 ton/min [20,28]. By comparison, the steel flow rate of the ultra-wide slab in this study was also at a relatively high level, but the maximum flow speed of the liquid surface (Figure 9) was significantly smaller than approximately 0.27–0.39 m/s, which is reported in the literature for conventional slabs. This further indicates that under the nozzle and mold width, because of ejection from the nozzle and the upper roll flow, the kinetic energy of the stream was greatly dissipated and moved to the meniscus, making the steel–slag interface lack enough dynamics during the final movement to the molten steel level.

Figure 10 shows the proportion of low-flow speed at different positions with different casting speeds. With increasing casting speed, the curve shows three gradient changes. When the casting speed was 0.9 m/min, the flow speed at each position of the whole liquid surface was more than 90% in the low-flow speed zone, and the activity of the liquid surface was very poor. With a quiet surface, slag entrainment was unlikely with this condition, but the meniscus might become too cold and stagnant. As the casting speed increased to 1.0–1.2 m/min, 85~95% of the surface flow speed was still in the low-flow speed zone, which was in the area of 180 mm near the nozzle and 90 mm near the narrow side. However, the liquid surface activity in the area 360 mm away from the center of the nozzle was obviously improved, and the proportion of the low-flow speed zone in this area decreased to 72.64~73.70%. The activity of the flow surface was significantly improved when the casting speed was 1.3 and 1.4 m/min, and the proportion of the low-flow speed area was significantly reduced. It decreased to 49.42% and 48.22%, respectively, near 360 mm from the center of the nozzle. In addition, the active region of the liquid surface at this flow speed has an obvious tendency to expand to the narrow surface. Therefore, increasing the casting speed could significantly improve the activity of the mold flow surface.

Due to the wide width and large width-to-thickness ratio, the main stream will collide with the broad surface first during the movement of the jet from the nozzle outlet to the narrow side, consuming a considerable amount of kinetic energy. This results in weak upper roll flow, slow renewal of high temperature molten steel at the steel–slag interface, and lack of sufficient activity, which was easy in producing slag strips and clusters, and which is harmful to the smooth CC operation and the surface quality of the strand. By optimizing the nozzle structure and adjusting the submerged depth to enhance the strength of the upper roll flow, the flow surface dynamics could be increased, thereby preventing the meniscus from becoming too cold or stagnant and improving the melting and flow of the mold flux.

## 4. Conclusions

The current study investigated the influence of casting speed on the fluid flow pattern and flow speed at the meniscus in an ultra-wide slab mold by using a water model. The main findings are as follows.

(1)For the ultra-wide slab, the ink tracing experiments indicate that there were various patterns existing in the flow field, and these patterns were constantly transforming into each other. The proportions of symmetric and asymmetric flows were 13~29% and 71~87%, respectively.(2)The angle of the jet was approximately twice of the nozzle downward angle. The jet momentum from the nozzle ports was diffused by colliding with the wide face surface, which lowered its kinetic energy and affected its subsequent diffusion.(3)The variation of surface flow speed with time near both sides of the nozzle or the narrow face was relatively symmetrical at a casting speed of 1.3 m/min. The most active area was located within 300–500 mm of the width center, and the flow speed was in the 0.14–0.23 m/s range.(4)At different casting speeds, the distribution of average surface flow speed was C-shaped. As the casting speed increased from 0.9 to 1.4 m/s, the corresponding peak increased from 0.08 to 0.2 m/s, and the proportion of flow with low speed gradually decreased.

## Figures and Tables

**Figure 1 materials-16-01135-f001:**
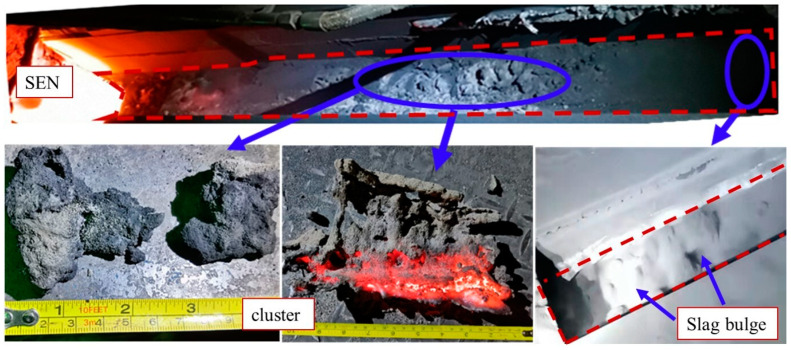
Top surface of slag pool in mold with a cross-section of 2920 × 150 mm.

**Figure 2 materials-16-01135-f002:**
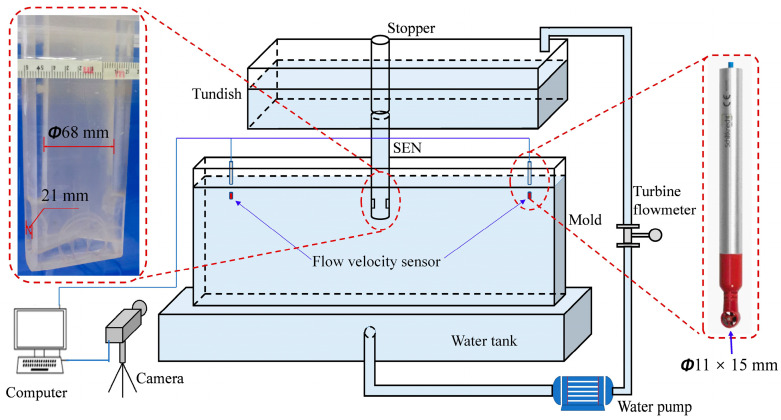
Schematic of water model in this study.

**Figure 3 materials-16-01135-f003:**
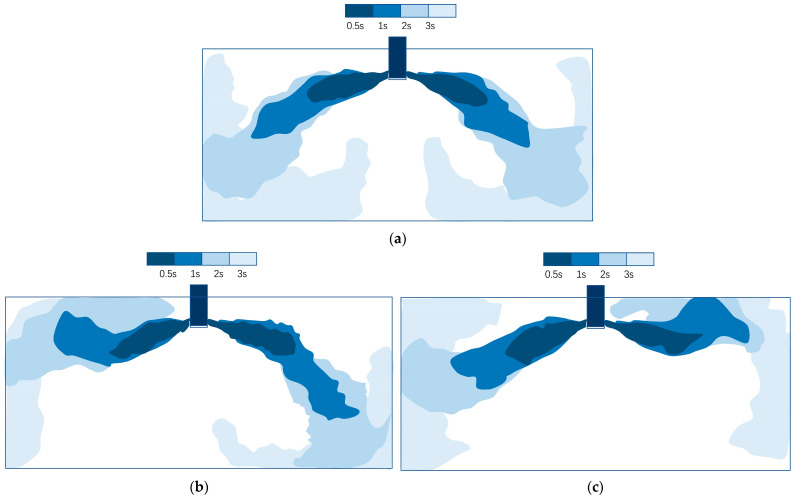
Typical evolution of fluid flow: (**a**) symmetrical (**b**) and (**c**) asymmetric patterns.

**Figure 4 materials-16-01135-f004:**
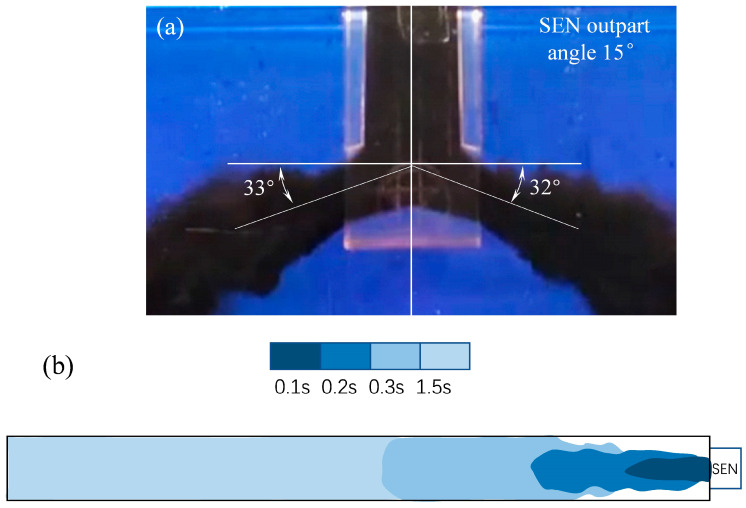
Schematic of (**a**) jet angle and (**b**) top view for the symmetrical flow.

**Figure 5 materials-16-01135-f005:**
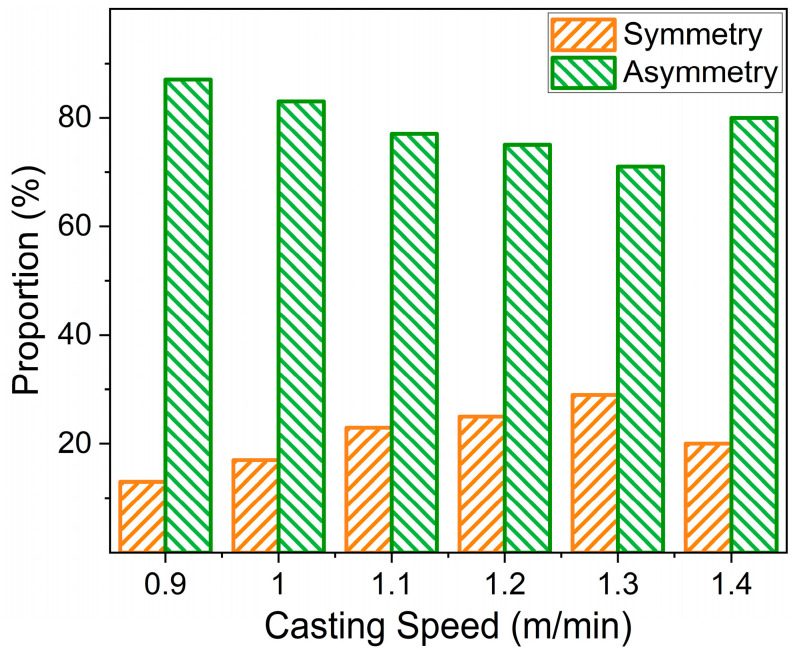
Proportion of symmetric and asymmetric flow patterns at different casting speeds.

**Figure 6 materials-16-01135-f006:**
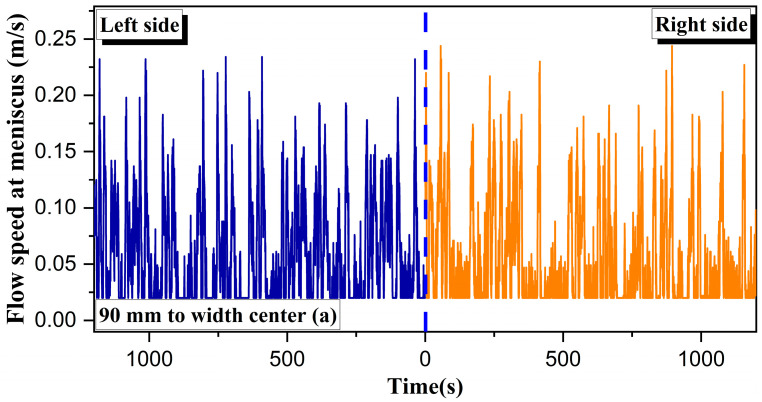
Flow speed beneath the meniscus at different positions from the width center: (**a**) 90 mm, (**b**) 360 mm, and (**c**) 630 mm.

**Figure 7 materials-16-01135-f007:**
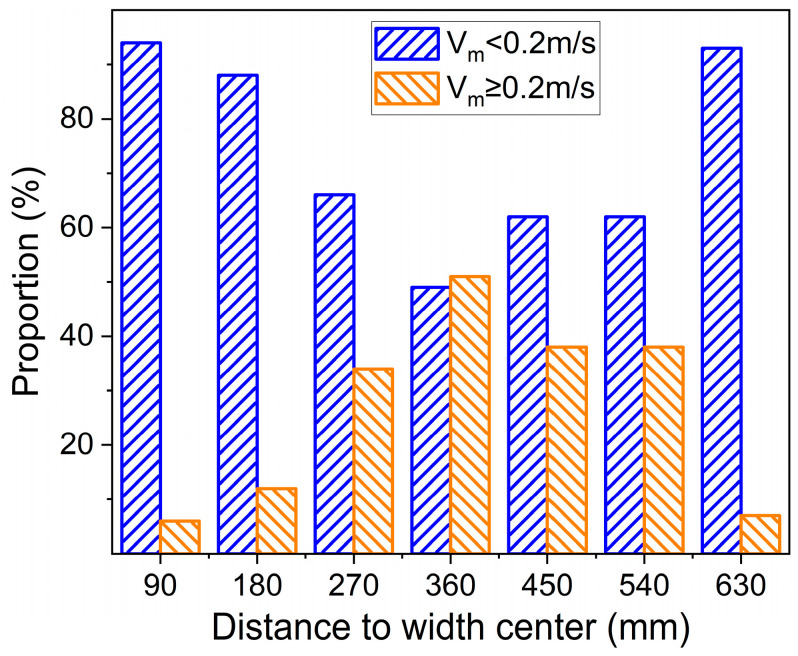
Proportion of speed range at different positions at the meniscus.

**Figure 8 materials-16-01135-f008:**
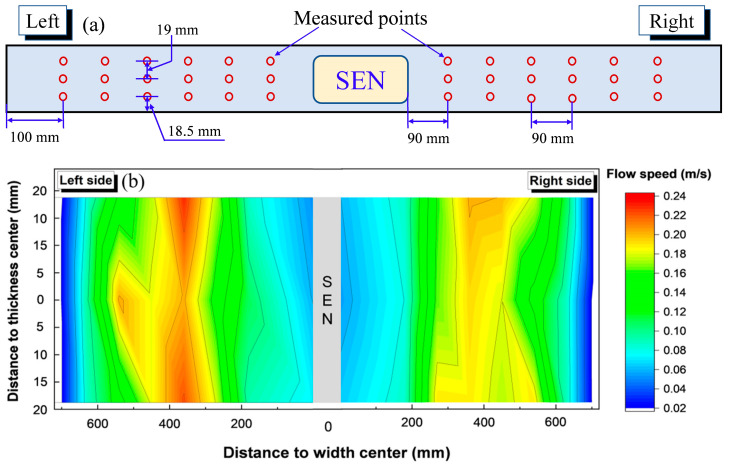
The distribution of the measurement locations (**a**) and the average surface flow speed (**b**).

**Figure 9 materials-16-01135-f009:**
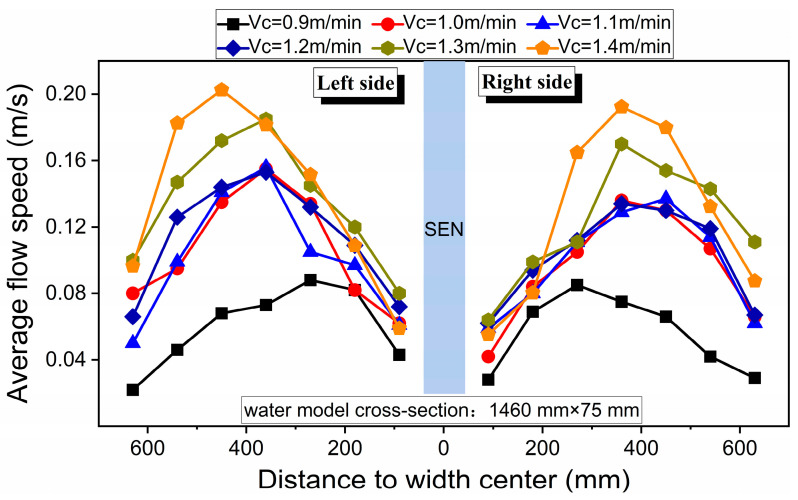
Influence of casting speed on the distribution of average flow speed at the meniscus.

**Figure 10 materials-16-01135-f010:**
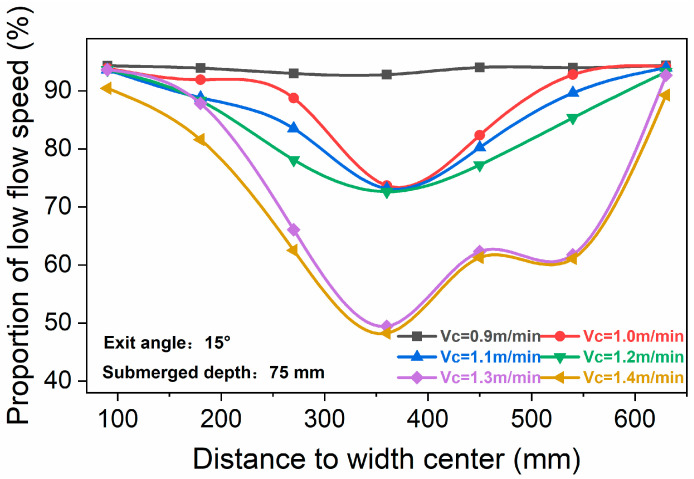
Proportion of low speed at different positions under different casting speeds.

**Table 1 materials-16-01135-t001:** Main parameters of the physical model and the prototype.

Parameter	Prototype	Model
Cross-section (mm × mm)	2920 × 150	1460 × 75
Nozzle exit (mm × mm)	42 × 95	21 × 47.5
Exit angle (°)	15	15
Submerged depth (mm)	150	75
Casting speed (m·min^−1^)	0.9–1.4	0.636–0.990

## Data Availability

Not applicable.

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
