# Peer review of "Fluid Flow in Continuous Casting Mold for Ultra-Wide Slab"

_materials, 2023, doi:10.3390/ma16031135_

Round 1

Reviewer 1 Report

The manuscript brings new knowledge in the field of evaluation and optimization of the continuous casting process - CC. It is easy to read and understand, so it has the potential to appeal to a wider public, not directly focused on the foundry industry.

From a scientific and industrial point of view, it has a high value, but it is necessary to eliminate the mentioned shortcomings.

Line 79 - 85: "The Materials and Methods should be described with sufficient details to allow oth ers to replicate and build on the published results. Please note that the publication of your manuscript implicates that you must make all materials, data, computer code, and protocols associated with the publication available to readers. Please disclose at the submission stage any restrictions on the availability of materials or information. New methods and protocols should be described in detail while well-established methods can be briefly described and appropriately cited." - A remnant from a flood template. Please remove it

Line 265: "This section is not mandatory but can be added to the manuscript if the discussion is  unusually long or complex." - A remnant from a flood template. Please remove it

Minor flaws in following the template. For example bullet points and spaces after image captions. Please edit.

The Discussion section is missing. Please add and discuss in detail

Reviewer 2 Report

The manuscript "Fluid flow in continuous casting mold for ultra-wide slab" describes the simulation of the molten steel flow during the casting of the ultrawide slab. The Simulation is made on the water model. The values of the material flow, as well as their symmetry/asymmetry are evaluated.

The reviewer has a few notes:

1. The L14 of the abstract not clear. Please rewrite it.

2. What is the source of Fig.1? Is it necessary to put it into the Introduction?

3. The first paragraph of Section 2 seems out of place and seems to be copied from the guide for Authors. Please check that.

4. Same goes for the Section 5

5. The Conclusions are too long - they should be significantly shortened.

6. From the conclusions it is unclear, what conditions the authors would recommend in case of casting the steel.

7. The quality of Fig. 9 and Fig. 10 should be improved.

The manuscript is recommended for the publication after minor revision.

Reviewer 3 Report

Line 58 to 59 starting in “The top….”: Why the top surface of the slag pool was more active near the narrow side. Using the images below how can we find out this conclusion? A better/clear explanation is necessary....

Line 79 to 85: I think that this part came from mdpi template. You must remove this....

Line 159: This figure suggests a speed with maximization of symmetric flow. What about include some discussion on this topic?

Line 168: Please, it looks obvious, but also, specify the position of the thickness where you got flow speeds.

Line 184 (about Figure 6.): Perhaps if you put this graphics using just only one chart will show better a comparative flow speed between the three different positions. By this way you must change flow speed scale.

Line 253 (about Figure 10): What is the limit to define "low flow rate"? or "low flow speed"? Both are the same subject? What is the set point to be considered as "low flow speed" or "low flow rate"?

In “x” axis: instead "Distance to width center", What about left just "Position width (mm)";

Line 265: You must to delete this part.

Round 2

Reviewer 1 Report

I thank the authors for taking my comments into account and editing the submitted manuscript

I think that it is possible to publish the manuscript in this form